# How Has the Treatment of Polish Children with Dravet Syndrome Changed? Future Perspectives

**DOI:** 10.3390/biomedicines12061249

**Published:** 2024-06-04

**Authors:** Anita Zielińska, Urszula Skarżyńska, Paulina Górka-Skoczylas, Tomasz Mazurczak, Aleksandra Kuźniar-Pałka, Karolina Kanabus, Dorota Hoffman-Zacharska, Elżbieta Stawicka

**Affiliations:** 1Clinic of Paediatric Neurology, Institute of Mother and Child, Kasprzaka 17A, 01-211 Warsaw, Poland; anilewandowska@imid.med.pl (A.Z.); tomasz.mazurczak@imid.med.pl (T.M.); alekuzniar@imid.med.pl (A.K.-P.); 2Department of Organization and Accounts, Institute of Mother and Child, Kasprzaka 17A, 01-211 Warsaw, Poland; urszula.skarzynska@imid.med.pl; 3Department of Medical Genetics, Institute of Mother and Child, Kasprzaka 17A, 01-211 Warsaw, Poland; paulina.gorka@imid.med.pl (P.G.-S.); karolina.kanabus@imid.med.pl (K.K.); dorota.hoffman@imid.med.pl (D.H.-Z.)

**Keywords:** Dravet syndrome, fenfluramine, stiripentol

## Abstract

Background: This report focuses on the treatment histories of 21 patients diagnosed with Dravet syndrome (DRVT) under the care of the Mother and Child Institute in Warsaw. This paper aims to present typical treatment schemes for patients with drug-resistant epilepsy, as well as to highlight the influence of genetic diagnosis on pharmacotherapeutic management and to present an economic analysis of hospitalization costs. This paper will also summarize the effectiveness of the latest drugs used in DRVT. Methods: Clinical data were collected retrospectively from available medical records. The effectiveness of anticonvulsant treatment was assessed based on epileptic seizure diaries and observations by caregivers and pediatric neurologists. Results: The study group (*n* = 21) consisted of patients aged 3–26 years. Orphan drugs dedicated to Dravet syndrome were introduced in all patients due to the genetic diagnosis, which significantly improved the patients’ clinical conditions. The breakthrough drugs were stiripentol (in 16/21) and fenfluramine (in 3/21). Conclusions: In recent years, molecular genetics has rapidly developed in Poland, along with a steady increase in knowledge of Dravet syndrome among the medical profession. Early and precise diagnosis provides the opportunity to target treatment with drugs dedicated to Dravet syndrome with high efficacy.

## 1. Introduction

Dravet syndrome (DRVT) is an infantile-onset developmental and epileptic encephalopathy associated with drug-resistant epilepsy (which occurs when a patient has failed to become (and stay) seizure free with adequate trials of two antiseizure medications) and comorbidities like intellectual disability, autistic features, behavioral problems, ataxia, and sleep disorders [1]. A total of 80% of all cases are due to pathogenic variants in the *SCN1A* gene. The genetic etiology was described in 2001. The syndrome usually appears in the first year of life in a previously healthy infant. Children have unilateral or bilateral febrile or febrile tonic–clonic seizures. The seizures often progress to status epilepticus (SE). Between 1 and 4 years of age, these seizures persist, and children develop other types of seizures, such as atypical absences and myoclonic seizures [2]. The frequency of DS is estimated to be 1:16,000 to 1:46,000 live births [3].

The treatment of patients with DRVT has remarkably changed in recent years. In 2011, the list of antiepileptic drugs was short: benzodiazepines, valproate, ethosuximide, and bromides [4]. The first breakthrough in pharmacological treatment was the introduction of stiripentol, which significantly reduced the rate of recurrent life-threatening status epilepticus. Drug-resistant status epilepticus is a significant problem for patients with DRVT, especially in childhood. Patients with Dravet syndrome also have an increased risk of premature death due to SUDEP (sudden unexpected death in epilepsy). Stiripentol is indicated for use in conjunction with clobazam and valproate as an adjunctive therapy for refractory generalized tonic–clonic seizures in patients with DRVT. The mechanism of action appears to increase brain levels of gamma-aminobutyric acid (GABA). This could occur by the inhibition of the synaptosomal uptake of GABA and/or the inhibition of GABA transaminase. Stiripentol has also been shown to enhance GABA-A receptor-mediated transmission in the immature rat hippocampus and increase the mean open duration (but not the frequency) of GABA-A receptor chloride channels using a barbiturate-like mechanism [4]. In recent years, the European Medicines Agency (EMA) has approved cannabidiol for treatment, and in 2023, it approved fenfluramine, which is already a second-line drug in the international consensus for the treatment of DRVT after valproic acid. The fenfluramine antiepileptic effect is associated with agonism at 5-HT1D, 5-HT2A, 5-HT2C, and 5-HT4 receptors and antagonism at 5-HT1A receptors. Further research into new drugs focuses on serotonergic transmission and treating the cause of the disease; clinical trials with antisense oligonucleotides and gene therapy are underway.

The authors describe the treatment histories of 21 patients under the care of the Mother and Child Institute in Warsaw from 2009 to 2024. The article aims to present typical treatment schemes for patients with drug-resistant epilepsy (whose epileptic seizures occur despite receiving two antiseizure drugs at the maximum allowed dose), as well as to highlight the influence of genetic diagnosis on pharmacotherapeutic management and to present an economic analysis of hospitalization costs. This paper will also summarize the effectiveness of the latest drugs used in DRVT.

## 2. Materials and Methods

Clinical data were collected retrospectively from available medical records, including medical observations, specialist consultations, laboratory and EEG (electroencephalography) results, and interviews with caregivers of our patients. The effectiveness of anticonvulsant treatment was assessed based on epileptic seizure diaries and observations by caregivers and pediatric neurologists. From 2009 until July 2023, we had 231 patients from centers nationwide in our genetic database with pathogenic mutations in the *SCN1A* gene; from our center, there were 78 with a confirmed mutation.

Patients were confirmed with a diagnosis of Dravet syndrome based on the following criteria (according to ILAE 2022 [5]):Onset of seizures typically between 3 and 9 months and, in rare cases, 1–20 months.Standard head size during the first years of life.Intellectual disability or regression of psychomotor development.Types of seizures: recurrent focal clonic (hemiclonic) febrile and afebrile seizures (which often alternate sides from seizure to seizure), focal to bilateral tonic–clonic, and/or generalized clonic seizuresAdditional seizure types (not mandatory): myoclonic seizures, focal impaired awareness seizures, focal to bilateral tonic–clonic seizures, atypical absence seizures, atonic seizures, nonconvulsive status epilepticus, tonic, and tonic–clonic seizures mainly in sleep and in clusters.Drug-resistant epilepsy.Course of illness: drug-resistant seizures with the potential of episodes of status epilepticus.

The exclusion criteria:Epileptic spasms.Early infantile *SCN1A* DEE.No history of prolonged seizures (>10 min).Lack of fever sensitivity as a seizure trigger.Typical EEG background without interictal discharges after the age of two years.Focal neurological findings.MRI showing a causal focal lesion: brain malformation, hypoxic-ischemic brain injury, brain tumor, neurocutaneous disorders, etc.First seizure diagnosis during the neonatal period.

For the study, we selected 21 patients who have been constantly under the care of our center since their genetic diagnosis, in whom we were able to precisely follow the treatment and the impact on the clinical condition.

Characterization of the study group was carried out using basic statistical methods (standard deviation, mean, student *t*-test, ANOVA, and standard deviation).

## 3. Results

The study group (*n* = 21) consisted of patients aged 3–26 years (data for 2024), with a mean age of 14 years (SD = 6.1). The M/F ratio was 10/11. Their clinical and molecular characteristics are presented in Table 1. Among the patient group, 1/21 did not meet the criteria for DRVT. The girl (P_1) had heterozygous deletions on chromosome 2, including all exons (Ex1_26del) of the *SCN1A* gene. She had a very severe course of the disease, was a recumbent patient, was profoundly disabled, was without the ability to communicate, and epileptic seizures appeared as early as three months of age. In the study group, the first epileptic seizure occurred in patients between 4 and 8 months of age (on average at five months); in 12/21, this was a tonic–clonic seizure, prolonging into status epilepticus, all provoked by hyperthermia. Others had focal seizures. Patients had a variety of seizure types: myoclonic (in younger patients), focal clonic, atypical absence seizures, and focal onset status epilepticus. In the last psychological examination, 3/21 had no intellectual disability, 5/21 had mild disability, 8 had moderate disability, 4 had severe disability, and 1 had profound intellectual disability. Only two patients had no difficulties with mobility.

Most (19/21) patients were initially treated with valproic acid, and two were treated with vigabatrin. Subsequently, due to a lack of improvement, other anticonvulsants were used (Figure 1). After receiving the result of the genetic test confirming the presence of *SCN1A* pathogenic variants, stiripentol, clobazam, and, in a few patients, bromide or fenfluramine were included in the treatment (Figure 1).

In three patients, significant clinical improvement was observed after fenfluramine was used as an added therapy (Figure 1). One patient took the drug for too short a time to see an effect on overall functioning. In the other two patients treated with fenfluramine, there was no deterioration of cognitive or executive functions, ataxia symptoms significantly decreased, and rapid speech development occurred.

Fenfluramine was introduced in three patients (P_7, P_18, P_21) from 2023–2024. Since 2023, the drug has been available and funded in Poland under the emergency access procedure. P_7 is a 4-year-old girl previously treated with a 2.5:1 ketogenic diet, stiripentol, clobazam, and valproic acid. Despite treatment, she presented with tonic–clonic seizures with a frequency of 1–2 times a week and myoclonic seizures, from a single one during the day to a series of them lasting up to 30 min. The sleep EEG recorded multiple multifocal and generalized discharges (groups and series lasting up to 6 s) of spikes/polyspikes, spike-and-wave discharges with an amplitude of up to 500 uV, and multifocal single, group, and sometimes series of sharp and slow wave complexes with an amplitude of 50–150 uV.

The patient P_7 started treatment with fenfluramine on November 2022. Starting at a dose of 0.4 mg/kg, a significant improvement in seizure control was observed (reduction over 70%), with tonic–clonic seizures occurring once every two weeks, myoclonus every four days, and an acceleration of the child’s developmental dynamics. Remarkably, the patient’s clinical improvement increased with the length of fenfluramine treatment. The EEG recording is currently normal; tonic–clonic seizures occur only during infections, are short and resolve spontaneously after about a minute, and the parents have not observed myoclonic seizures for six months.

P_18 is currently a 7-year-old boy who, before the use of potassium bromide, had epileptic seizures at a frequency of about once a month, requiring ICU (intensive care unit) hospitalization. After the addition of potassium bromide in 2022, a significant reduction in the number of seizures was observed, but tonic–clonic seizures provoked by hyperthermia continued, requiring emergency benzodiazepines. Before adding fenfluramine, the patient was treated with potassium bromide, topiramate, valproic acid, and stiripentol. The patient received fenfluramine in February 2024 and is currently on a dose of 0.4 mg/kg body weight. During the 2-month follow-up period, one tonic–clonic seizure occurred (during the introduction of the drug, not on the current dose). Since the symptoms of ataxia, lethargy, and feelings of weakness increased, according to his parents, the dose of stiripentol was recently reduced.

Patient P_21: Before the inclusion of fenfluramine treatment, tonic–clonic seizures occurred on average once every 25 days, with a seizure duration of several seconds to several minutes, often requiring interruption with benzodiazepines. The patient was treated with valproic acid, topiramate, stiripentol, and levetiracetam. In EEG during sleep, very numerous generalized discharges (groups and series) consisting of sharp waves and sharp wave–slow wave complexes with an amplitude of up to 900 uV, localized in the temporal and central–parietal–occipital areas (on one side or both sides), were reported. Fenfluramine treatment was introduced on September 2023 as an add-on drug. At a dose of 0.4 mg/kg, topiramate discontinuation was started. The parents observed a decreased appetite in the child but no weight loss. An improvement in sleep and a reduction in ataxia symptoms were observed. As of March 2023, the girl remained seizure-free; the control EEG recording showed no abnormalities.

P_9 is a 15-year-old patient treated with valproic acid, topiramate, levetiracetam, and additional stiripentol. Despite ongoing polytherapy, he continued to present with multiple epileptic seizures, often progressing to status epilepticus tonic–clonic seizures requiring intravenous benzodiazepines. In 2020, potassium bromide was added to the treatment. The seizures were significantly reduced in frequency, occurred only with infections, and no status epilepticus was observed. In 2023, the number of seizures increased. They began to appear on average once every two weeks, prolonged. In EEG recordings during sleep, sharp waves, spikes/polyspikes, and sharp wave–slow wave discharges with an amplitude of 100–600 uV were observed. The patient was eligible for rescue access to fenfluramine, but due to the abnormal echocardiography required for eligibility, mitral valve leaflet prolapse and suspected hypertrophic cardiomyopathy, the decision to disqualify was made, and cannabidiol was added to the treatment. The patient has been receiving the drug since February 2024. Initially, he experienced an allergic reaction in the form of a rash, but the drug was continued, and his parents observed a slight reduction in seizures. The patient has so far used all available antiepileptic drugs dedicated to Dravet syndrome. The next step in his treatment is to try to add cenobamate and implant a vagus nerve stimulator.

The cost analysis:

The analysis of the cost of treating patients with DRVT highlights the highest diagnostic expenditures in overall hospital expenses. In our study group, they accounted for about 36.61% and 35.20% of total treatment costs (Table 2).

When converted to euros, the example laboratory cost of P_7’s two hospitalizations was about 461.50 euros out of 1260.62 euros; for P_21, it was about 668.31 euros out of 1898.54 euros. The profitability between total costs and revenues was negative.

## 4. Discussion

In 80% of patients, the first drug used was valproic acid. In the period before the diagnosis of DRVT, patients were also treated with other antiepileptic drugs, such as levetiracetam, vigabatrin, phenobarbital, and topiramate. Only 24% (5 patients) achieved satisfactory clinical improvement. All patients in this group were treated with valproic acid, and four were additionally treated with a benzodiazepine preparation, four with topiramate, and one with levetiracetam. Most of them did not achieve a significant reduction in seizures, which is consistent with the descriptions of such a group of patients observed in literature [6,7]. A total of 28% of patients were also receiving drugs such as lamotrigine or carbamazepine, with significant clinical decline, which is also consistent with available scientific data [8,9]. After the diagnosis of Dravet syndrome, stiripentol was added to the treatment in 76% of patients, 50% of whom showed significant improvement, defined as a >50% seizure reduction. The remaining patients continued to have frequent epileptic seizures, problems with sleep, aggressive behavior, and ataxia. The literature data report a wide range of response rates, from 23% to 84%, in all types of seizures in DRVT [9,10]. Many authors explain this wide range of clinical responses by analyzing the reduction of different types of epileptic seizures, as well as the influence of the patient’s age and previous treatment [11]. In many studies, the most significant efficacy is observed in younger children, where stiripentol was used promptly [12,13]. In our study group, we were unable to perform a reliable statistical analysis due to the small number of patients (*n* = 21). However, polytherapy with stiripentol added to valproic acid was most effective in the youngest children, particularly in regard to the reduction of status epilepticus, as is reported in other studies [9,10].

In three patients, a significant seizure reduction was observed after the addition of bromide to stiripentol. According to literature, bromide does not have strong recommendations [4,5,14]. Bromide is not available in many countries. In Poland, it is registered for the treatment of drug-resistant epilepsy.

Studies show that fenfluramine is a highly effective drug in reducing epileptic seizures and is safe for use even in the youngest patients [15,16,17].

According to Bishop [18], the addition of fenfluramine to treatment, especially in children during the period of the most intense psychomotor development (under 5 years of age), has a significant effect on improving behavior, cognitive functioning, and emotion regulation, which is not directly related to the reduction of epileptic seizures. Still, it is due to the drug’s mechanism of action itself. In children with DRVT, this may be of particular importance in the context of the invalidating course of the disease [19].

The most recent drug available for the treatment of seizures with focal onset is cenobamate. Focal seizures often occur in patients with Dravet syndrome. The mechanism of action of cenobamate is unknown. Perhaps cenobamate acts on GABA-A receptors as a positive allosteric modulator, but it also blocks persistent sodium currents by promoting inactive states of sodium channels. Sodium channel blockers are prohibited drugs in the treatment of patients with DS. Nevertheless, four adult patients with Dravet syndrome treated with cenobamate with great success are described in the literature [20]. All mutation variants in the *SCN1A* gene were described as LoF. The follow-up time of patients previously treated with multiple antiepileptic drugs was between 300–540 days. All patients achieved a sustained seizure reduction of more than 80%, which is exceptional for such drug-resistant epilepsy. The introduction of this drug should be considered in the absence of fenfluramine treatment, as in the case of P_9.

A future treatment opportunity for DS patients is personalized therapy, taking into account the type of mutation and genotype–phenotype correlations [21]. In our study group, in patient P_18, no significant differences were observed in terms of clinical course or response to treatment. However, due to the pathogenic variant that causes gain of function (GOF), the inclusion of treatment with sodium channel blockers is something to consider in the future.

Another direction of current research is to develop causative treatment targeting the lack of NaV1 sodium channel alpha subunit protein deficiency [22]. Studies have been conducted in mouse models using antisense nucleotide (ASO) molecules that increase the expression of a productive transcript of the *SCN1A* gene [23]. The application of ASO on days 2 or 14 after birth was shown to significantly reduce the frequency of seizures and decrease the risk of SUDEP.

It is also noteworthy that before genetic diagnosis and targeted therapy, parents of children with DRVT may be excluded from professional productivity due to demanding and time-consuming child care. Such exclusion generates indirect costs that can be comparable to direct medical expenses [24]

In the light of the studies [25,26], the costs associated with diagnosis and treatment both before and after a DRVT diagnosis identified in Poland appear to be lower compared to international data. This may be due to the differences in prices for medical services and the healthcare management system in Poland. The negative profitability between total costs and revenues underscores the health system’s challenge in financing the treatment of this rare but cost-intensive condition. Ultimately, the economic data clearly show that treating Dravet syndrome requires significant investment in medical and non-medical resources, which directly affects hospital budget management. In a long-term context, investments in genetic diagnostics and novel therapies are expected to help reduce the need for hospitalization, especially for emergency reasons, potentially saving money for the hospital. However, these are long-term projections, and currently, high investments may only pay off after a more extended period.

## 5. Conclusions

In recent years, molecular genetics has rapidly developed in Poland, along with a steady increase in knowledge of Dravet syndrome among the medical profession. This allows for an increasingly early diagnosis of DRVT, often as early as the first year of life. Early and precise diagnosis provides the opportunity to target treatment with drugs dedicated to Dravet syndrome with high efficacy. This impacts both the patient’s condition and improves the child’s quality of life, as well as reduces the impact of the disease on the parents’ working lives and lowers the indirect social and economic costs associated with care.

The treatment of Dravet syndrome is associated with high diagnostic costs, placing a significant economic burden on the healthcare system. At the same time, the prevalence and reduction in the price of genetic testing offer improved patient outcomes and the possibility of ultimately reducing costs. High initial expenses currently pose a dilemma for medical administrators. Reconciling the financial requirements (associated with high-cost genetic diagnostics) with the anticipated benefits requires the consideration of both the economic consequences and the importance of pioneering DRVT therapies.

Drugs that have been introduced in recent years, such as fenfluramine, cannabidiol, or cenobamate, represent a breakthrough in treatment for many patients, due to their high efficacy in reducing the number of epileptic attacks. In addition, due to its unique mechanism of action, fenfluramine appears to have a positive effect on the development of cognitive and executive functions, significantly improving the prognosis of DRVT.

## Figures and Tables

**Figure 1 biomedicines-12-01249-f001:**
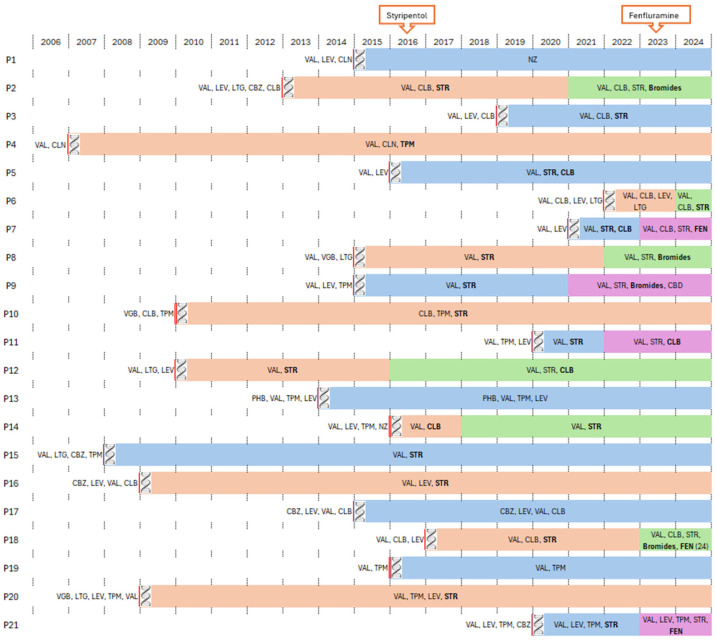
Drugs used in patients before and after genetic diagnosis was established. Stiripentol has been available in Poland since 2016, fenfluramine since 2023. VAL—valproic acid, CLB—clobazam, LEV—levetiracetam, TPM—topiramate, CLN—clonazepam, CBZ—carbamazepine, LTG—lamotrygine, STR—stiripnetol, FEN—fenfluramine, NZ—nitrazepam, PHB—phenobarbital, VGB—vigabatrine, Bromides—potassium bromide, CBD—Cannabidiol. 
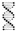
—the date of genetic diagnosis. Bold represents drugs with significant efficacy.

**Table 1 biomedicines-12-01249-t001:** Clinical and molecular characteristics of the group.

ID	Age	Sex	DRVT Criteria ^1^	Intellectual Disability	Motor Dysfunction	*SCN1A* Mutation (Nav1.1 Protein; c.DNA)	Inheritance	Variant Pathogenicity ^2^
P_1	9	F	no	profound	does not sit, walk	Ex1_26del	de novo	LOF
P_2	15	M	yes	moderate	ataxia	p.Lys1846Serfs * 11 (c.5536_5539del)	de novo	LOF
P_3	7	M	yes	severe	ataxia	p.Thr808Hisfs * 29 (c.2420dup)	de novo	LOF
P_4	17	F	yes	mild	ataxia	p.Arg1596His (c.4787G>A)	paternal	Neut/prob. LOF *
P_5	14	F	yes	mild	motor clumsiness	p.Arg1636 * (c.4906C>T)	de novo	LOF
P_6	4	F	yes	normal	none	p.Arg1927Gly (c.5779A>G)	de novo	Pat/LOF
P_7	3	F	yes	mild	ataxia	p.Arg101Gln (c.302G>A)	de novo	Neut/prob. LOF *
P_8	22	F	yes	moderate	ataxia	p.Ile227Ser (c.680T>G)	de novo	Pat/LOF
P_9	9	M	yes	moderate	ataxia	p.Gly1655Val (c.4964G>T)	de novo	Pat/LOF
P_10	16	F	yes	mild	ataxia	p.Arg931Cys. (c.2791C>T)	de novo	Pat/LOF
P_11	4	M	yes	normal	motor clumsiness	p.Leu1425 * (c.4274T.A)	de novo	LOF
P_12	26	F	yes	severe	motor clumsiness	p.Asn1487Profs * 22 (c.4459_4460del)	de novo	LOF
P_13	26	F	yes	moderate	ataxia	p.Lys1846Serfs * 11 (c.5536_5539del)	de novo	LOF
P_14	8	M	yes	moderate	reduced tension	p.? (c.2947-1G>A)	de novo	LOF
P_15	20	F	yes	severe	ataxia	p.Val143fs * 148 (c.429_430 del)	nd	LOF
P_16	24	M	yes	mild	motor clumsiness	p.? (c.3421_3429+7del)	de novo	LOF
P_17	18	M	yes	moderate	none	p.Ala342Val (c.1025C>T)	de novo	Neut/prob. LOF *
P_18	7	M	yes	moderate	ataxia	p.Asn416Ser. (c.1247A>G)	de novo	Pat/GOF
P_19	17	M	yes	moderate	ataxia	p.Phe1710Ser (c.5129T>C)	de novo	Pat/LOF
P_20	24	M	yes	severe	motor clumsiness	p.Arg580 * (c.1738C>T)	de novo	LOF
P_21	5	F	yes	normal	none	p.Arg1927Gly. (c.5779A>G)	de novo	Pat/LOF

^1^ Dravet syndrome criteria according to ILAE 2021 [5]. ^2^ Pathogenicity and predicted functional consequences calculated using the machine learning model funNcion, a functional variant prediction in Navs and Cavs ion channels (https://funnc.shinyapps.io/shinyappweb/, accessed on 30 March 2024).

**Table 2 biomedicines-12-01249-t002:** Analysis of hospitalization costs for patients with Dravet syndrome.

	P_21 Hospitalization from 6 to 9 October 2019	P_21 Hospitalization from 24 to 27 June 2019	P_7 Hospitalization from 21 to 24 April 2021	P_7 Hospitalization from 20 to 27 May 2021	Average Costs
Length of hospitalization	3	3	3	7	
Medicines	€29.71	€29.88	€65.61	€54.47	€44.92
Disposable materials	€34.05	€59.52	€35.33	€77.32	€51.56
Laboratory tests	€461.50	€36.01	€668.31	€79.14	€311.24
Imaging tests	€32.35	€125.08	€75.48	€202.71	€108.90
Personnel costs	€501.80	€644.55	€891.94	€2251.36	€1072.41
Non-medical costs	€200.56	€209.88	€161.22	€944.94	€379.15
Total Costs	€1260.61	€1105.56	€1898.54	€3611.46	€1968.18
Total Revenues	€759.94	€738.64	€1217.07	€850.22	
Profitability	−€500.02	−€366.27	−€680.83	−€2759.73	
Genetic tests during hospitalization	YES	NO	YES	NO	

## Data Availability

The data presented in this study are available upon request from the corresponding author.

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
