# Peer review of "How Has the Treatment of Polish Children with Dravet Syndrome Changed? Future Perspectives"

_biomedicines, 2024, doi:10.3390/biomedicines12061249_

Round 1
Reviewer 1 Report
Comments and Suggestions for Authors
Dear Authors,
Having reviewed your manuscript, I have several concerns that warrant attention. Firstly, while the introduction provides some information, there are notable gaps that hinder clarity and completeness. Secondly, the Materials and Methods section lacks detail, particularly concerning inclusion and exclusion criteria, which are fundamental for understanding the study's methodology. Furthermore, the presentation of results is unclear, and the scientific style throughout the manuscript requires improvement. Additionally, the small sample size (n=21) and the absence of statistical analyses undermine the robustness of the findings. Lastly, the presentation of results on a case-by-case basis makes generalization challenging, considering the unique characteristics of each patient.
and those are specific comments:
abstract
In the abstract please remove in background ‘’ the authors descirbve’’ rephrase the sentence
Also the word article
Correct please ‘’ and lead pediatric neurologists.””
Line 20 , correct N=20 by n=20
Introduction
How can you explain the drug resistancy while it is an infantile diseance?
Line 35 and 37 references are needed
Also in introduction you can give more details about DRTB such as the prevalence of the disease…
Line 44 and 53 start by stating the all word than the abbreviation
Line 59, it will be better if you remove ‘’ In this article, 59 the authors describe’
Materials and methods
EEg please write it in all
The M&M section it too poor, add details about how data collection also how many patients had the didease from 2009 to 2024 also inclusion and exclusion criteria
Results
In line 76 what do you mean by 2024 data
Line 86 patiets had instead of have
Line 91 is not understood
Line 103 correct available
Line 113 correct
Line 119 which patient ?
Line 120 significatant improvement how that was assessed ?
Line 123-125 correct
ICU write it inall
Line 169 highest instead of high
Discussion
Line 206-210 to be in result section not discussion
Conclusions are too long
Comments on the Quality of English Language
Please correct the English conjugation in some sentences; the present tense is incorrect while the past tense is correct. Additionally, some words are misspelled, and some sentences are not clear
Author Response
Thank you.
We have corrected as you recommended. Please see the attachment

Reviewer 2 Report
Comments and Suggestions for Authors
The design of this paper is relatively reasonable and reliable.They describe the treatment history of 21 patients diagnosed with Dravet Syndrome (DRVT) under the care of the Mother and Child Institute in Warsaw.present typical treatment schemes for patients with drug-resistant epilepsy, as well as to highlight the influence of genetic diagnosis on pharmacotherapeutic management and an economic analysis of hospitalization costs. The assignment was provided that early and precise diagnosis provides the opportunity to target treatment with drugs dedicated to Dravet syndrome with high efficacy. The article has pro relativity effect than academic power. However, there are still many issues that need to be carefully revised in this thesis.
1. The chosen of volunteers were random? What is the under-control situation? Pleas add it.
2. Does the situation of Personnel costs more over than medicines fee was reasonable?
3. Please provide the evidence of improving the prognosis of DRVT.
4. Please taking grammar check.
Comments on the Quality of English LanguageThere are too many Polish local English grammar.
Author Response
1. The chosen of volunteers were random? What is the under-control situation?
Please add it.
Patients in the study group were not randomly selected. They were patients with
Dravet syndrome in whom we had the most accurate data on molecular diagnosis
and drug treatment history. At the same time, patients who were treated with the
latest drugs such as fenfluramine were selected and described in more detail.
2. Does the situation of Personnel costs more over than medicines fee was
reasonable?
In answering the question of whether the higher medical staff costs relative to drug
costs are justified, it is important to note several key aspects related to the
introduction of fenfluramine in Poland. Due to the relatively short period of its use in
the Polish health system, it is difficult at this stage to clearly assess the long-term
financial effectiveness of the use of this drug, especially in the context of a potential
reduction in hospitalizations.
The lack of long-term data makes it impossible to fully analyze the benefits that the
public payer could gain from a reduction in hospitalizations. Theoretically, the
reduction in seizure frequency and severity that the drug could provide could lead to
significant savings over time, placing less burden on the health system by reducing
the need for intensive hospital care and associated costs.
In this situation, although medical staffing costs may initially outweigh spending on
the drug, such an investment may be justified in anticipation of long-term effects that
will reduce the overall cost of treating and caring for patients with Dravet syndrome.
However, this requires ongoing evaluation and monitoring of the effectiveness of the
therapy to accurately assess its cost-effectiveness in the future.
2. Please provide the evidence of improving the prognosis of DRVT.
At this stage, there are no randomized, prospective studies analyzing the
effectiveness of fenfluramine treatment. This is an issue that requires further
analysis. However, we do have data confirming an improvement in the prognosis and
course of Dravet syndrome in patients treated with stiripentol, particularly in
combination with valproic acid and clobazam. Studies show the >50% reduction in
the frequency of clonic or tonic-clonic seizures during the second month of the
double-blind period compared to baseline, were 71% on the stiripentol arm and 5%
on placebo. The second study showed similar results (67% of responders on
stiripentol vs. 9% on placebo).
ï‚· Chiron C. Stiripentol for the treatment of seizures associated with Dravet
syndrome. Expert Rev. Neurother. 2019;19:301–310. doi:
10.1080/14737175.2019.1593142. [PubMed] [CrossRef] [Google Scholar]
ï‚· Chiron C., Marchand M.C., Tran A., Rey E., D’Athis P., Vincent J., Dulac O.,
Pons G., STICLO Study Group Stiripentol in severe myoclonic epilepsy in
infancy: A randomised placebo-controlled syndrome-dedicated trial. Lancet.
2000;356:1638–1642. doi: 10.1016/S0140-6736(00)03157-3.
Reviewer 3 Report
Comments and Suggestions for Authors
General Comments:
The article explores advancements in treating Dravet Syndrome in Poland, emphasizing the role of orphan drugs and genetic diagnostics in improving patient outcomes and reducing costs. It provides useful insights but needs substantial improvement to meet the journal’s standards. I recommend a major revision.
Major Points:
1. Innovativeness: The study reiterates known treatments without introducing new insights. Incorporating innovative treatment strategies or recent advancements would enhance its value.
2. Comparative Analysis: The lack of comparison with other treatment options weakens the manuscript. Adding a control group or comparative data is essential for validating the claimed efficacy of the treatments discussed.
3. Economic Analysis: The economic analysis needs to be expanded to include a detailed cost-effectiveness comparison with other treatments and long-term financial impacts.
4. Statistical Rigor: The manuscript should detail the statistical methods used to strengthen the credibility of the results.
5. The conclusion could be more impactful with tighter language and a focus on the core findings. I recommend condensing it to enhance clarity and readability.
Author Response
1. Innovativeness: The study reiterates known treatments without introducing new
insights. Incorporating innovative treatment strategies or recent advancements
would enhance its value.
The study, in addition to describing standard methods of treating patients, takes into
account the use of the most recent substances available on the market, for example,
fenfluramine. The center's experience, as the first in Poland to treat patients with
Dravet syndrome, is innovative and unique in the country. In addition, the authors
described the most up-to-date methods of treatment, including the planned
introduction of antisense oligonucleotide therapy.
2. Comparative Analysis: The lack of comparison with other treatment options
weakens the manuscript. Adding a control group or comparative data is
essential for validating the claimed efficacy of the treatments discussed.
We agree that the lack of a control group is a limitation of our study. However, the
main purpose of our study was not to examine the efficacy of treatment in Dravet
syndrome, but rather to present the history of pharmacotherapy, highlight the impact
of genetic diagnosis on management and prognosis, and share recent experiences
through descriptions of patients treated with fenfluramine in Poland.
3. Economic Analysis: The economic analysis needs to be expanded to include a
detailed cost-effectiveness comparison with other treatments and long-term financial
impacts.
I agree that it is important to expand the economic analysis, with a comparison of the
cost-effectiveness of different treatments for Dravet syndrome. However, it should be
noted that we have currently no data on the long-term financial impact of these
therapies. Fenfluramine was administered for the first time in Poland at the Mother
and Child Institute in August 2023, meaning that this was the first patient in Poland to
receive the drug. As a result, we lack data to fully assess the long-term impact of this
therapy on treatment costs in the context of the Polish healthcare system.
Fenfluramine is funded under the Rescue Access to Drug Technology (RDTL), which
allows patients to access this therapy in critical situations when other treatment
options have failed or are not available. Epidiolex, on the other hand, has been
funded under the drug program since April 17, 2024, providing long-term support for
patients who may benefit from this therapy as part of their regularly funded care.
Such funding options are key to reducing the financial burden on patients and their
families while improving access to modern and effective therapies. An economic
analysis that takes these aspects into account would allow for a more complete
understanding of the economic benefits of different treatment options. This is a
significant area for future research.
3. Statistical Rigor: The manuscript should detail the statistical methods used to
strengthen the credibility of the results.
Characterization of the study group was carried out using basic statistical methods
(standard deviation, mean, student's T test, Anova, standard deviation,)
Round 2
Reviewer 1 Report
Comments and Suggestions for Authors
Dear authors in the attached file you sent I didn't find the author response
plase send it
Comments on the Quality of English Languagesome sentences are difficult to understand please correct
Author Response
Thank you, we did it.
All comments have been considered, and adequate corrections were made in the text. We would like to express our special gratitude for the careful evaluation and detailed analysis of our work.
Reviewer 3 Report
Comments and Suggestions for Authors
Thank you for thoughtfully responding and revising the manuscript
Author Response
Thank you
Round 3
Reviewer 1 Report
Comments and Suggestions for Authors
dear authors some abbreviations in the text still undefined ex ILAE
M&M must be following STROBE
in M&M authors declared using statistical methods such as SD, ANOVA , T test but not of those were used in results section
also the study sample n=21, make impossible to be representative
in general the paper doesn't have a clear scientific style of writing, introduction also discussion